# The trunk replaces the longer mandible as the main feeding organ in elephant evolution

Chunxiao Li[1,2], Tao Deng[1,2]*, Yang Wang[3], Fajun Sun[4], Burt Wolff[3], Qigao Jiangzuo[2], Jiao Ma[2], Luda Xing[1,2], Jiao Fu[1,2], Ji Zhang[5,6]*, Shiqi Wang[2]*

[1]University of Chinese Academy of Sciences, Beijing, China; [2]Key Laboratory of Vertebrate Evolution and Human Origins of the Chinese Academy of Sciences, Institute of Vertebrate Paleontology and Paleoanthropology, Chinese Academy of Sciences, Beijing, China; [3]Department of Earth, Ocean and Atmospheric Science, Florida State University, Tallahassee, United States; [4]Environmental Science & Technology, University of Maryland, College Park, United States; [5]School of Civil and Hydraulic Engineering, Huazhong University of Science and Technology, Wuhan, China; [6]National Center of Technology Innovation for Digital Construction, Wuhan, China

*For correspondence:
dengtao@ivpp.ac.cn (TD);
zhang_ji@hust.edu.cn (JZ);
wangshiqi@ivpp.ac.cn (SW)

Competing interest: The authors declare that no competing interests exist.

**Abstract** The long-trunked elephantids underwent a significant evolutionary stage characterized by an exceptionally elongated mandible. The initial elongation and subsequent regression of the long mandible, along with its co-evolution with the trunk, present an intriguing issue that remains incompletely understood. Through comparative functional and eco-morphological investigations, as well as feeding preference analysis, we reconstructed the feeding behavior of major groups of longirostrine elephantiforms. In the *Platybelodon* clade, the rapid evolutionary changes observed in the narial region, strongly correlated with mandible and tusk characteristics, suggest a crucial evolutionary transition where feeding function shifted from the mandible to the trunk, allowing proboscideans to expand their niches to more open regions. This functional shift further resulted in elephantids relying solely on their trunks for feeding. Our research provides insights into how unique environmental pressures shape the extreme evolution of organs, particularly in large mammals that developed various peculiar adaptations during the late Cenozoic global cooling trends.

## eLife assessment

This study presents **fundamental** findings on the evolution of extremely elongated mandibular symphysis and tusks in longirostrine gomphotheres from the Early and Middle Miocene of northern China. The integration of multiple methods provides **compelling** results in the eco-morphology, behavioral ecology, and co-evolutionary biology of these taxa. In doing so, the authors elucidate the diversification of fossil proboscideans and their likely evolutionary responses to late Cenozoic global climatic changes.

## Introduction

Proboscideans are known for their exceptionally elongated and versatile trunks (*Shoshani, 1998*). However, unlike modern elephants, proboscideans underwent a prolonged evolutionary phase characterized by the presence of greatly elongated mandibular symphysis and mandibular tusks. This elongation can be traced back to the Late Oligocene species *Palaeomastodon* and *Phiomia*, which

**eLife digest** The elephant's trunk is one of the most efficient food-gathering organs in the animal kingdom. From large branches to thin blades of grass, it can coil around and bring many types of vegetation to the animals' strong, short mandibles. This versatility allows elephants to thrive in a range of environments, including grasslands.

Trunks are not the only spectacular feature to emerge in Proboscideans, the family of which elephants are the only surviving group. During the early and middle Miocene (between 23 to 11.6 million years ago), many of these species had dramatically elongated lower jaws; how and why this trait emerged then disappeared is poorly understood. The role that lengthened mandibles and trunks played during feeding also remains unclear.

To address these questions, Li et al. focused on *Platybelodon*, *Choerolophodon* and *Gomphotherium*, which belong to three Proboscidean families that roamed Northern China between 17 and 15 million years ago. Each had elongated lower jaws, but with strikingly distinct lengths and morphologies.

Chemical analyses on enamel samples helped determine which habitat the families occupied, while mathematical modelling revealed how their mandibles tackled different types of plants. Trunk shape was assessed via analyses of the nasal region.

The results suggest that *Choerolophodon* had mandibles better suited for processing branches and a short, 'primitive' trunk. *Gomphotherium* sported a versatile jaw that could handle both grass and trees, as well as a rather 'elephant-like' trunk. The jaw of *Platybelodon* seemed well-adapted to cut grass, and remarkable bone structures point towards a long, strong and flexible trunk. While modern elephants fully depend on their trunks to eat, morphological constraints suggest that, in these species, the appendage only served to assist feeding (e.g., by pressing down on branches).

All families shared an environment that included grasslands and forests, but analyses suggest that, for a period, *Choerolophodon* favored relatively closed habitats while *Platybelodon* spread into grasslands and *Gomphotherium* navigated both landscapes. This suggests that the evolution of long, strong and flexible trunks is tightly associated with grazing.

About 14 million years ago, a global cooling event led to grasslands expanding worldwide. The fossil record shows the mandibles of Proboscideans starting to shorten after this period, including in the descendants of *Gomphotherium* that would give rise to modern elephants. The work by Li et al. sheds light onto these evolutionary processes, and the environmental pressures which helped shape the trunk.

are among the earliest elephantiforms (*Andrews, 1906*), and continued through to the Late Miocene *Stegotetrabelodon*, a stem elephantid (*Shoshani, 1996*; *Tassy, 1996*). Extreme longirostriny, a feature observed in fossil and modern fishes, reptiles, and birds, was relatively rare among terrestrial mammals and its occurrence in large-bodied proboscideans is particularly intriguing. Particularly, during the Early and Middle Miocene (approximately 20–11 Ma), the morphology of mandibular symphysis and tusks exhibited remarkable diversity, with over 20 genera from six families (Deinotheriidae, Mammutidae, Stegodontidae, 'Gomphotheriidae', Amebelodontidae, and Choerolophodontidae) displaying variations (*Shoshani, 1996*; *Tassy, 1996*). Why did proboscideans have evolved such a long mandible of so diversified morphology? How did fossil proboscideans use their strange mandibular symphysis and tusks, and what was the role of trunk in their feeding behavior? Finally, what was the environmental background for the co-evolution of their mandible and trunks, and why did proboscideans finally lose their long mandible? These important issuers on proboscideans evolution and adaptation remain poorly understood. Addressing these significant aspects of proboscidean evolution and adaptation requires comprehensive investigations into the functional and eco-morphology of longirostrine proboscideans.

## Results
### Mandible morphology of bunodont elephantiforms
During the Early and Middle Miocene, bunodont elephantiforms, as the ancestral group of living elephants, flourished (*Osborn, 1936*; *Gheerbrant and Tassy, 2009*; *Cantalapiedra et al., 2021*).

Bunodont elephantiforms include Amebelodontidae, Choerolophodontidae, and 'Gomphotheriidae'; all possess a greatly elongated mandibular symphysis (*Gheerbrant and Tassy, 2009*; *Cantalapiedra et al., 2021*). Our comprehensive phylogenetic reconstructions contained the majority of longirostrine elephantiform taxa at the species level and strongly supported this taxonomic scheme (*Figure 1A*; *Figure 1—figure supplement 1*, *Figure 1—source data 1*). The three families were characterized by their distinctive mandible and mandibular tusk morphology (*Shoshani, 1996*; *Tassy, 1996*). The paraphyletic 'Gomphotheriidae' have clubbed lower tusks (*Figure 1B and K*; *Figure 1—figure supplement 2B*); their mandibular symphysis is relatively narrow. This morphology is rather unspecialized, and the extant elephants are derived from 'Gomphotheriidae' (*Tassy, 1996*; *Cantalapiedra et al., 2021*). The mandibular symphysis of Amebelodontidae is generally shovel-like and the mandibular tusks are usually flattened and wide. *Platybelodon* is the most specialized genus within this family; it possesses extremely flattened and widened mandibular tusks with a sharp distal cutting edge (*Wang et al., 2013*: *Figure 1C, I, and J*; *Figure 1—figure supplement 2A*). Choerolophodontidae is unique because it completely lacks mandibular tusks and it has a long trough-like mandibular symphysis (*Figure 1D, L, and M*, *Figure 1—figure supplement 2C–F*). A very deep slit is present on each side of the distal alveolar crest (distal mandibular trough edge: *Figure 1L*), which is presumably for holding a keratinous cutting plate (*Figure 1M*), similar to the slits for holding large claws in felids and some burrowing mammals (e.g. anteaters). The anterior mandibular foramen of *Choerolophodon* is extremely large and tube-like (*Figure 1—figure supplement 2D*), which indicates a very developed mental nerve and eponymous artery (*Eales, 1926*), for the nutrition of tissues (i.e. keratins) grown from symphyseal dermis. The three lineages exhibit different evolutionary states of food acquisition organs (PC1 scores from mandible and tusk characters; see Materials and methods: *Figure 1—figure supplement 4*; *Figure 1—figure supplement 5B*). Different food acquisition organs morphology strongly indicate different methods of food acquisition among the three gomphothere families.

## Evolutionary dynamics and ecological niches of three longirostrine bunodont elephantiforms families

The three longirostrine bunodont elephantiforms families were widely distributed in the Early–Middle Miocene in northern China, ranging from ~19 to 11.5 Ma (*Figure 2*, *Figure 2—figure supplement 1*) (most of the Shanwangian and Tunggurian stages of the Chinese Land Mammal Age) (*Wu et al., 2018*). However, the relative abundances of these three families were very different and varied through time. We investigated fossil bunodont elephantiforms from four regions: Linxia Basin (LX), Tongxin region (TX), Junggar Basin (JG), and Tunggur region (TG) (*Figure 2*, *Figure 2—figure supplement 1*, *Figure 2—source data 1*). We focused on different fossil assemblages in different ages. We evaluated the amounts of fossils of each elephantiform taxon from major fossil assemblages in the above regions from three museums (IVPP, HPM, and AMNH, for full names, please see the Materials and methods) and calculated their proportions among all bunodont elephantiforms fossils (*Figure 2A*; *Figure 2—figure supplement 2A*). During ~19–17 Ma, the quantities of both *Choerolophodon* (Choerolophodontidae) and *Protanancus* (Amebelodontidae) were larger than those of *Gomphotherium* ('Gomphotheriidae') despite the high diversity of *Gomphotherium* in species level. During ~17–15 Ma (the Mid–Miocene Climate Optimum [MMCO]), the relative fossil abundances of the three gomphothere families were similar. However, within Amebelodontidae, *Platybelodon* appeared and rapidly replaced the primitive *Protanancus*. After ~14.5 Ma (i.e. the beginning of the Mid–Miocene Climate Transition [MMCT]) (*Westerhold et al., 2020*), *Choerolophodon* suddenly experienced regional extinction, and *Gomphotherium* also gradually declined. Within Amebelodontidae, *Aphanobelodon*, which is morphologically similar to *Platybelodon*, only occurred in one fossil assemblage, and the *Platybelodon* population greatly expanded. After ~13 Ma, *Gomphotherium* were regionally extinct, and *Platybelodon* became the only genus among bunodont elephantiforms.

We analyzed stable isotope values of the tooth enamel for each gomphothere taxon in different fossil assemblages (*Figure 2—source data 2*). Overall, the $\delta^{13}$C values indicate a relatively open environment that consisted of a diverse range of habitats (including grasslands, wooded grasslands, and forests) dominated by $C_3$ plants in northern China from ~19 to 11.5 Ma (*Figure 2B*, *Figure 2—figure supplement 1*). In LX1 (*Figure 2B*; *Figure 2—source data 2*), at approximately ~18.5 Ma (see *Supplementary file 1*), both $\delta^{13}$C and $\delta^{18}$O values of *Choerolophodon* and *Protanancus* are similar and have a wide range of overlap, and those of *Gomphotherium* are in the middle of the range.

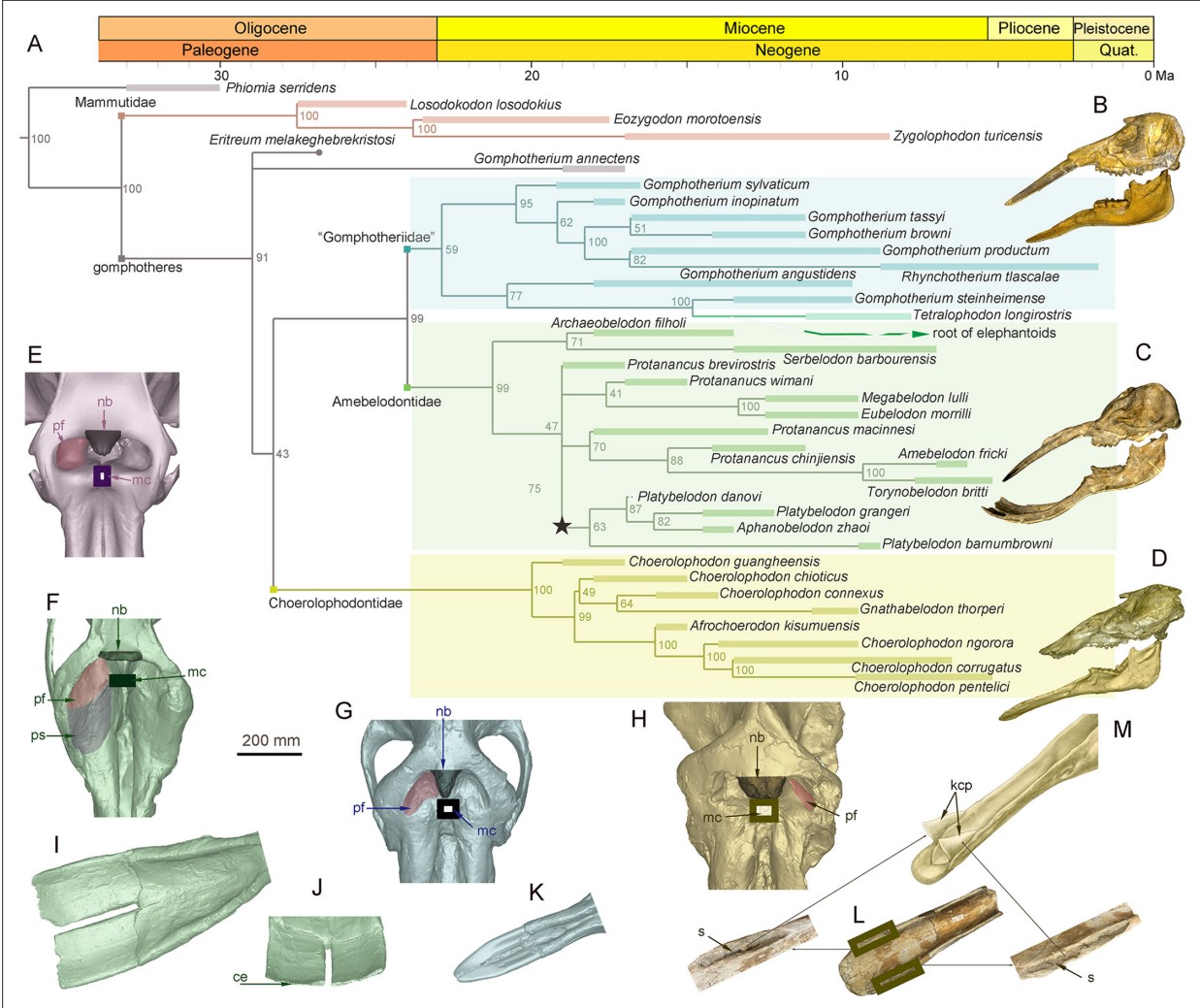

**Figure 1.** Morphology of the narial region and mandible of three gomphothere families compared with an extant elephant, and the elephantiformes phylogeny. (**A**) Phylogenetic reconstruction of major longirostrine elephantiforms at the species level based on the Bayesian tip-dating method. The node support (the number at each node) is the posterior probability, and the bars represent chronologic ranges of each taxon. (**B–D**) Representative cranium and mandible specimens of the three gomphothere families, including IVPP V22780, cranium, and IVPP V22781, mandible, of *Gomphotherium tassyi* [B], 'Gomphotheriidae', from Heijiagou Fauna, Tongxin region (TX4); HMV 0930, cranium and associated mandible of *Platybelodon grangeri* [C], from Zengjia Fauna, Linxia Basin (LX3); and IVPP V23457, cranium and associated mandible of *Choerolophodon chioticus* [D], from Middle Miaoerling Fauna, Linxia Basin (LX2). (**E–H**) Narial morphology of bunodont elephantiforms and elephantids in dorsal view, including IVPP OV733, *Elephas maximus* [E], a living elephantid; HMV 0930, *P. grangeri* [F]; IVPP V22780, *G. tassyi* [G]; and IVPP V23457, *C. chioticus* [H]. Mandibular morphology of bunodont elephantiforms. (**I–J**) Mandibular symphysis and tusks of HMV 0930, *P. grangeri*, in dorsal [I] and distal [J] views. (**K**) Mandibular symphysis and tusks of IVPP V22781, *G. tassyi*, in dorsal view. (**L**) Mandibular symphysis of IVPP V25397, *C. chioticus*, showing the deep slits at both sides of the distal alveolar crests in dorsal view. (**M**) Reconstruction of keratinous cutting plates in the slits, in dorsolateral view. Anatomic abbreviations: ce, cutting edge of the distal mandibular tusk in *Platybelodon*; kcp, reconstructed keratinous cutting plates in *Choerolophodon*; nb, nasal process of nasal bone; mc, slit or groove for mesethmoid cartilage insertion (white in color); pf, perinasal fossa; ps, prenasal slope in *Platybelodon*; s, slit for holding kcp in *Choerolophodon*.

The online version of this article includes the following source data and figure supplement(s) for figure 1:

**Source data 1.** Morphological characters and data set for phylogenetic analyses, characters, see Appendix.

**Figure supplement 1.** Phylogenetic reconstruction of longirostrine bunodont elephantiforms and mammutids using the maximum parsimony method (tree length, 253; CI, 0.470; RI, 0.779).

**Figure supplement 2.** Mandible of longirostrine bunodont elephantiforms.

**Figure supplement 3.** Narial region of longirostrine bunodont elephantiforms.

**Figure supplement 4.** Evolutionary levels of narial region (**A**) and of characters in relation to horizontal cutting (**B**).

**Figure supplement 5.** Character combines in relation to feeding behavior in trilophodont longirostrine bunodont elephantiforms.

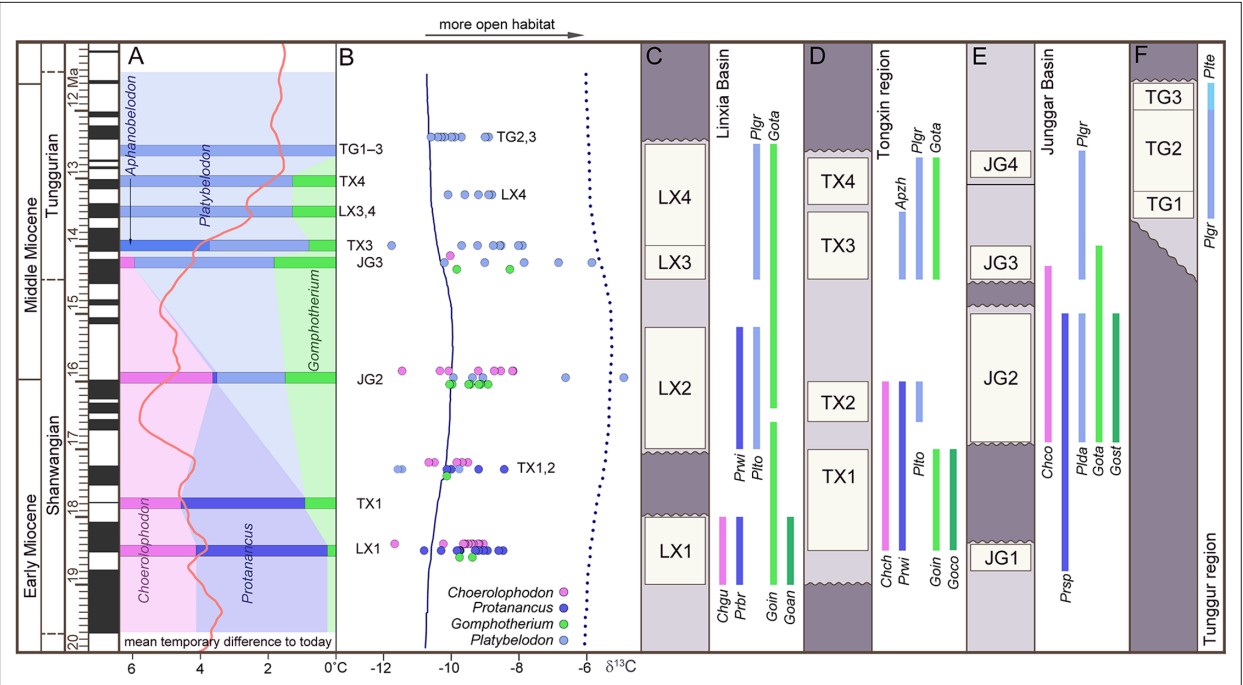

**Figure 2.** Relative abundance, tooth enamel δ13C, and stratigraphic ranges of the three gomphothere families in the Shanwangian and Tunggurian stages (~20–11.5 Ma) of northern China. (**A**) Relative abundance of the three gomphothere families, including Choerolophodontidae, only represented by *Choerolophodon* (pink); Amebelodontidae, represented by *Protanancus*, *Aphanobelodon* (dark blue), and *Platybelodon* (light blue); and 'Gomphotheriidae', only represented by *Gomphotherium* (green). Horizontal bars indicate the average ages of the fossil assemblages, which are shown in **C–F**. The ages were determined by paleomagnetism (**Supplementary file 1**). The red curve shows the global reference benthic foraminifer oxygen isotope curve, which represents the global temperature (after **Westerhold et al., 2020**). (**B**) Tooth enamel stable carbon isotopic compositions of various gomphothere taxa. Each circle represents the bulk enamel δ13C values of a single tooth. The data of LX4 and TG2,3 are from previous publications (**Wang and Deng, 2005**; **Zhang et al., 2009**). Solid and dashed lines represent the mean and maximum enamel δ13C values for C3 diets that have been corrected for Miocene atmospheric $CO_2$ δ13C (after **Tipple et al., 2010**). (**C–F**) Synthetic stratigraphic columns of typical fossil bearing regions of northern China during ~19–11.5 Ma, which incorporated different fossil assemblages with different ages, from the Linxia Basin [**C**], Tongxin region [**D**], Junggar Basin [**E**], and Tunggur region [**F**]. Vertical bars represent the temporal ranges of different gomphothere taxa. Abbreviations for gomphothere taxa: Apzh, *A. zhaoi*; Chch, *C. chioticus*; Chco, *C. connexus*; Chgu, *C. guangheensis*; Goan, *G.* cf. *angustidens*; Goco, *Gomphotherium cooperi*; Goin, *G. inopinatum*; Gost, *G. steinheimense*; Gota, *G. tassyi*; Plda, *Platybelodon dangheensis*; Plgr, *P. grangeri*; Plte, *P. tetralophus*; Plto, *P. tongxinensis*; Prbr, *P. brevirostris*; Prsp, *Protanancus* sp.; Prwi, *P. wimani*.

The online version of this article includes the following source data and figure supplement(s) for figure 2:

**Source data 1.** Elephantiform specimens counted in the present article are housed in Institute of Vertebrate Paleontology and Paleoanthropology (IVPP), Hezheng Paleozoological Museum (HPM), and American Museum of Natural History (AMNH).

**Source data 2.** Original measurements of elephantiform tooth enamel isotope ratios analyses in the present article.

**Figure supplement 1.** Relative abundance and tooth enamel δ18O values of fossil elephantiforms from the Shanwangian and Tunggurian stages (~19–11.5 Ma) in northern China.

**Figure supplement 2.** Geographic distribution of *Platybelodon* worldwide.

This indicates that the niches of these three groups overlapped without obvious differentiation. In TX1,2 (~17.3 Ma) (**Deng et al., 2019**), during which *Platybelodon* first appeared, the δ13C values of *Platybelodon* are lower than those of *Protanancus*. The δ13C values of *Choerolophodon* and *Gomphotherium*, however, are all within the δ13C range of Amebelodontidae. In JG2 (~16 Ma), the δ13C value of *Platybelodon* shows a distinct positive shift, which indicates expansion into more open habitats (**Figure 2B**). However, *Choerolophodon* may have persisted in a relatively closed environment (**Figure 2B**). Moreover, the ecological niche of *Gomphotherium* appeared to be in between those of *Platybelodon* and *Choerolophodon*, and they potentially lived in the boundary area of open and closed habitats (**Figure 2B**). This isotopic niche pattern is also observed in JG3 (~14.3 Ma), with the rise of *Platybelodon* and decline of *Choerolophodon* and *Gomphotherium*. In TX3, LX4, and TG2,3 (after ~14 Ma), only *Platybelodon* were examined (as *Gomphotherium* specimens are too rare

to sample); its ecological niche was similar to that previously occupied by *Choerolophodon* after the latter went extinct.

## Specialized feeding behaviors of three gomphothere families

To reconstruct the feeding behaviors of the three gomphothere families, we performed finite element (FE) analyses on three models as representatives of each family: *Choerolophodon*, *Gomphotherium*, and *Platybelodon* (for model settings, see *Figure 3—figure supplements 1–4*; *Supplementary files 1 and 2*). We conducted two kinds of tests: the distal forces test and the twig-cutting test. In the distal forces test, after full muscle forces were exerted, a 5000 N vertical force is loaded on the distal end of each mandible (*Figure 3A–C*, *Videos 1–3*). The mandible strain energy curve (MSEC) of *Platybelodon* suddenly reaches a very high value, while the MSEC increases of *Gomphotherium* and *Choerolophodon* are far less than that of *Platybelodon* (*Figure 3D*). Then, keeping the magnitude of the external force unchanged, with change of the external force from vertical to horizontal direction, the MSEC of *Platybelodon* decreases even lower than that of *Gomphotherium* and was similar to that of *Choerolophodon*. This result indicates that the mechanical performance of the *Platybelodon* mandible is disadvantageous under distal vertical external forces, but greatly improved under horizontal external forces (*Figure 3A–C*, *Videos 1–3*).

In twig-cutting tests, a cylindrical twig model of orthotropic elastoplasticity was posed in three directions to the distal end of the mandibular tusks of *Platybelodon* and *Gomphotherium*, and to the keratinous cutting plate of *Choerolophodon*. The sum of the equivalent plastic strain (SEPS) from total twig elements was calculated (equivalent plastic strain represents the irreversible deformation of an element, and the sum from all twig elements can reflect the cutting effects) (*Figure 3E*), and the cutting videos are provided (*Videos 4–9*; *Video 10*; *Video 11*; *Video 12*; *Video 13*; *Video 14*; *Video 15*; *Video 16*; *Video 17*; *Video 18 Video 19*). When the twig was placed horizontally, the SEPS of the *Choerolophodon* model is the largest, which means that *Choerolophodon* has the highest twig-cutting efficiency, followed by *Gomphotherium*; while *Platybelodon* exhibits the much lower efficiency in cutting horizontal twigs than the other two models. When the twig was placed obliquely (45° orientation), the *Platybelodon* model still shows the smallest SEPS value, although it was nearly one order of magnitude higher than that of the horizontal twig. Finally, when the twig itself was in a vertical direction, the growth direction of the keratinous cutting plate determines that *Choerolophodon* cannot cut in this condition. The SEPS of the *Gomphotherium* model also decreases and shows lower cutting efficiency. In contrast, the SEPS of the *Platybelodon* model increases another order of magnitude, which is substantially larger than that of any other taxa in any cutting state. These data strongly indicate that *Platybelodon* mandible is specialized for cutting vertically growing plants. However, the *Choerolophodon* mandible is specialized for cutting horizontally or obliquely growing plants, this explains the absence of mandibular tusks, but they are likely not able to feed on vertically growing plants. The cutting effect of the *Gomphotherium* mandible is relatively even for all directions.

## Co-evolution of narial morphology and characters of horizontal cutting among gomphothere families

The evolutionary level of the trunk can be completely inferred from the morphology of the narial region (*Tassy, 1994*). Here, we first showed the narial region of a living elephant (*E. maximus*, IVPP OV733) (*Figure 1E*), focusing on the following four morphological factors. (1) The dorsal border of the narial aperture is slightly caudal to the postorbital process. This part is for the attachment of *maxilla-labialis*, which is the key muscle for manipulating the entire trunk (*Boas and Paulli, 1908*; *Eales, 1926*). (2) The narial aperture is wide, showing a pair of deep and sub-circular perinasal fossae. This part related to the insertion of *lateralis nasi*, and functions in enlarging the nostril cavity to suck up water in the trunk (*Boas and Paulli, 1908*; *Eales, 1926*). (3) The nasal process of the nasal bone is moderately developed. (4) The insertion slit for the mesethmoid cartilage is narrow and small, and deeply concealed in the narial aperture. The latter two points might be related to trunk flexibility. The smaller the nasal bone process and mesethmoid cartilage, the more flexible the trunk. These points were carefully discussed by *Tassy, 1994*.

The three lineages with different mandibular morphology also exhibited different stages of trunk evolution, which can be inferred from the narial region morphology (*Tassy, 1994*). Among the three groups, *Gomphotherium* has similar narial morphology to living elephantids (*Figure 1E and G*, *Figure 1—figure*

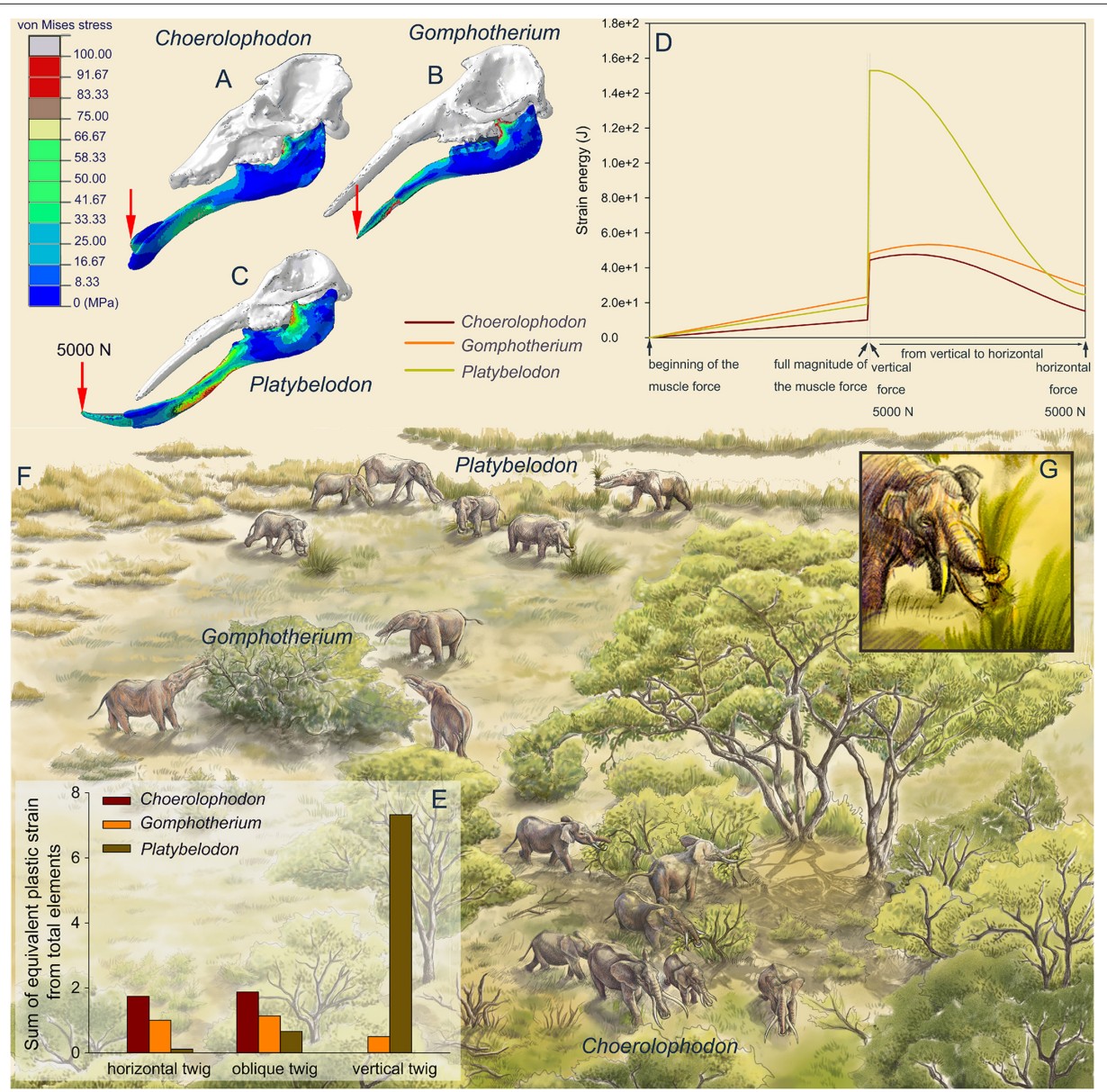

**Figure 3.** Finite element analyses of feeding behaviors among three longirostrine gomphothere families and reconstruction of their feeding ecology. (**A–C**) von Mises stress color maps of *Choerolophodon*, *Gomphotherium*, and *Platybelodon* models, with the full muscle forces exerted, and an additional 5000 N external vertical force applied on the distal end of the mandibular symphysis. (**D**) Strain energy curves of the three mandibles under the following three steps: (1) muscle forces linearly exerted; (2) a 5000 N external vertical force suddenly applied on the distal end; and (3) the 5000 N external force gradually changed from vertical to horizontal. (**E**) Sum of equivalent plastic strain from total elements (SEPS) of twigs cut by mandible models in three different directions (i.e. twig horizontal, 45° oblique, and vertical). Larger SEPS values indicate higher efficiency of twig cutting. (**F**) Scenery reconstruction of feeding behaviors of the three longirostrine gomphothere families (by X Guo), represented by *Choerolophodon* (Choerolophodontidae), feeding in a closed forest, *Gomphotherium* ('Gomphotheriidae'), feeding at the margin between the closed forest and open grassland, and *Platybelodon* (Amebelodontidae), feeding on open grassland. (**G**) Detailed 3D reconstruction of *Platybelodon* feeding by grasping the grass blades using their flexible trunk and cutting the grass blades using the distal edge of their mandibular tusks.

The online version of this article includes the following figure supplement(s) for figure 3:

**Figure supplement 1.** Geometric skull models for finite element (FE) analysis of three representative bunodont elephantiforms, including *Gomphotherium* (**A**), *Choerolophodon* (**B**), and *Platybelodon* (**C**), generated by Materialise 3-matic Research (V12.0).

**Figure supplement 2.** Mechanical settings of *Platybelodon* horizontal twig-cutting modeling; the blue cylinder represents the twig model.

**Figure supplement 3.** Mechanical settings of *Gomphotherium* vertical twig-cutting modeling.

**Figure supplement 4.** Mechanical settings of *Choerolophodon* oblique twig-cutting modeling.

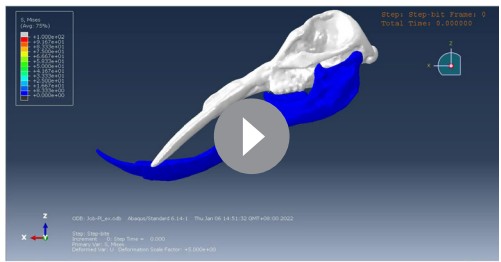

**Video 1.** Finite element (FE) modeling of *Platybelodon* distal force test, color map showing the von Mises stress.

https://elifesciences.org/articles/90908/figures#video1

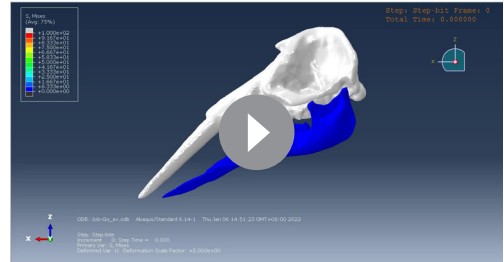

**Video 2.** Finite element (FE) modeling of *Gomphotherium* distal force test, color map showing the von Mises stress.

https://elifesciences.org/articles/90908/figures#video2

supplement 3D, E), with comparable size and morphology of the nasal bone process, insertion for mesethmoid cartilage, and perinasal fossae. Alternatively, the narial region of *Choerolophodon* shows a relatively primitive evolutionary stage (*Tassy, 1994*; *Figure 1H*, *Figure 1—figure supplement 3A–C*). It has a very wide and large nasal bone process for an elephantiform, a fairly wide and long groove for mesethmoid cartilage insertion, and a pair of somewhat incipient (or even absent) perinasal fossae. Amebelodontids usually possess common narial morphology similar to that of *Gomphotherium* (*Tassy, 1994*). However, *Platybelodon* has noteworthy differences in narial morphology (*Wang and Li, 2022*; *Figure 1F*, *Figure 1—figure supplement 3G–L*). Its nasal aperture is greatly enlarged, which results in a very broad area for attaching *maxillo-labialis* (*Boas and Paulli, 1908*; *Eales, 1926*), the 'core' muscle of the proboscis. The nasal bone process is very short and stout, with a slight dorsal bulge. The slit for mesethmoid cartilage insertion is very tiny. Beside the well-developed perinasal fossa, a vast inclined region is positioned rostral to the perinasal fossa, this is hereafter referred to as the prenasal slope (*Wang and Li, 2022*), and it potentially provides additional attachment for the *nasialis*.

The narial morphology of *Platybelodon* and the closely related genus *Aphanobelodon* (*Wang et al., 2017a*: *Figure 1—figure supplement 3F*), is unique in elephantiforms, and shows even more derived characters than living elephants. Principal components analysis (PCA) was respectively performed for the characters of the narial region and the food acquisition organs (mandible and tusks) to extract their synthetic characters. On the phylogenetic tree (*Figure 1—figure supplement 4A*), the *Platybelodon* lineage (the dark purple star) displays the most derived narial morphological combination (PC1 scores from characters in relation to the narial region), while Choerolophodontidae (the dark purple circle) show the least specialized narial morphology, which is close to the stem taxon *Phiomia*. Interestedly, the evolutionary level of the character combine in relation to horizontal cutting (in terms of PC2) is highly correlated with that of narial region (*Figure 1—figure supplement 4B*). In the *Platybelodon* clade, this character combine also shows high evolutionary level; while in the Choerolophodontidae clade, the evolutionary level of horizontal cutting is rather low, comparable with that of narial region. The result strongly suggests a highly co-evolution between narial region and horizontal cutting behavior in the trilophodont longirostrine bunodont elephantiforms.

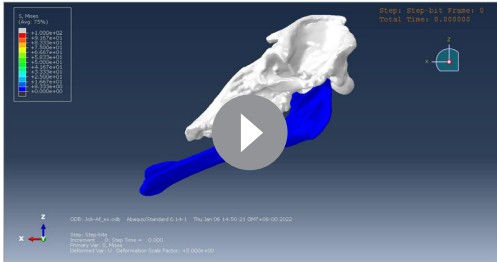

**Video 3.** Finite element (FE) modeling of *Choerolophodon* distal force test, color map showing the von Mises stress.

https://elifesciences.org/articles/90908/figures#video3

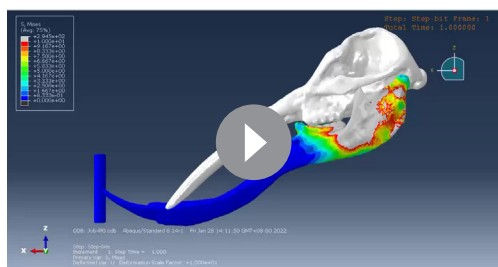

**Video 4.** Finite element (FE) modeling of *Platybelodon* vertical twig-cutting test, the total model, color map showing the von Mises stress.

https://elifesciences.org/articles/90908/figures#video4

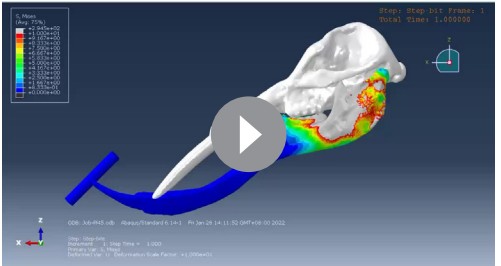

**Video 5.** Finite element (FE) modeling of *Platybelodon* vertical twig-cutting test, the twig model (facing to the surface that is in contact with the tusk), color map showing the equivalent plastic strain.

https://elifesciences.org/articles/90908/figures#video5

**Video 6.** Finite element (FE) modeling of *Platybelodon* oblique twig-cutting test, the total model, color map showing the von Mises stress.

https://elifesciences.org/articles/90908/figures#video6

## Discussion

In several fossil and living terrestrial mammalian groups, including living *Tapirus* (*Moyano and Giannini, 2017*) and extinct *Proboscidipparion* (*Ma et al., 2023*), *Astrapotherium* (*Kramarz et al., 2019*), and *Macrauchenia* (*Blanco et al., 2021*), elongated noses have evolved as food procuring organs. However, none of these groups possess a long and dexterous trunk like elephants, as they have not lost their mandibular incisors. Only living elephants are capable of solely accomplishing food procurement with their trunks. Coincidentally, longirostrine proboscideans, characterized by their extremely elongated mandibular symphysis and tusks, are the only group where these highly developed organs may have co-evolve (*Tassy, 1996*). It is evident that the great elongation of the nose is well matched with the extreme elongation of the mandibular symphysis. However, different lineages exhibit different evolutionary strategies strongly influenced by their ecological adaptations.

Our focus is on the ancestral elephantids, longirostrine bunodont elephantiforms, which include three families (Amebelodontidae, Choerolophodontidae, and 'Gomphotheriidae'). These groups flourished during the MMCO, a period of global warmth from 17 to 15 Ma (*Zachos et al., 2001*; *Westerhold et al., 2020*). With the climatic shift during the MMCT (after ~14.5 Ma), *Choerolophodon* sharply declined and experienced regional extinction in northern China. *Gomphotherium* also declined but persisted until around 13 Ma, while *Platybelodon* became predominant until the end of the Middle Miocene (*Figure 2A*, *Figure 2—figure supplement 1A*). Worldwide, *Choerolophodon* and *Gomphotherium* continued to flourish during the MMCT and survived to the early Tortonian in other regions (*Lambert and Shoshani, 1998*; *Göhlich, 1999*; *Sanders et al., 2010*). However, *Platybelodon* was mostly restricted to Central Asia, especially northern China, with only a few records found in other regions (*Wang et al., 2013*; *Wang and Li, 2022*: *Figure 2—figure supplement 2*). These uneven distributions were likely influenced by climatic and ecological factors, such as the relatively drier climate and more open ecosystems in northern China, and Central Asia, strongly affected by the elevation of the Tibetan Plateau (*Zachos et al., 2001*; *Wu et al., 2022*). Extensive studies,

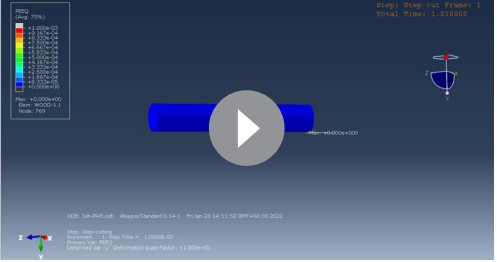

**Video 7.** Finite element (FE) modeling of *Platybelodon* oblique twig-cutting test, the twig model (facing to the surface that is in contact with the tusk), color map showing the equivalent plastic strain.

https://elifesciences.org/articles/90908/figures#video7

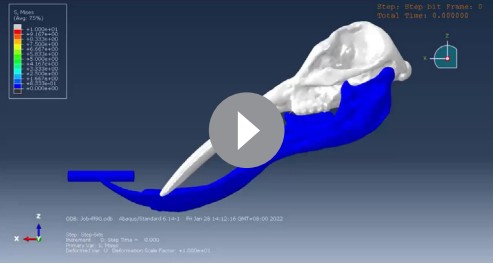

**Video 8.** Finite element (FE) modeling of *Platybelodon* horizontal twig-cutting test, the total model, color map showing the von Mises stress.

https://elifesciences.org/articles/90908/figures#video8

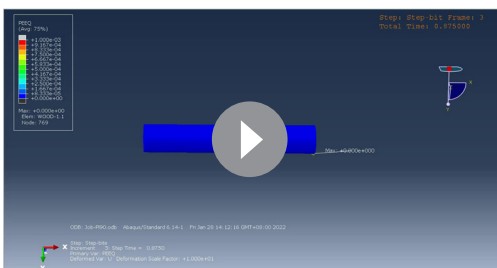

**Video 9.** Finite element (FE) modeling of *Platybelodon* horizontal twig-cutting test, the twig model (facing to the surface that is in contact with the tusk), color map showing the equivalent plastic strain.
https://elifesciences.org/articles/90908/figures#video9

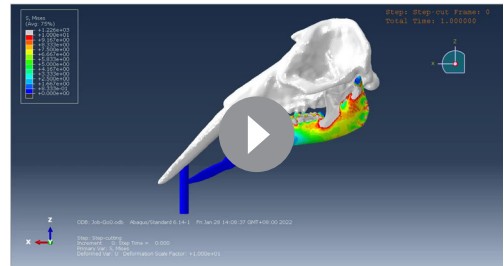

**Video 10.** Finite element (FE) modeling of *Gomphotherium* vertical twig-cutting test, the total model, color map showing the von Mises stress.
https://elifesciences.org/articles/90908/figures#video10

using evidence from various disciplines including geochemical and geomagnetic proxies, and fossil records, have been conducted to explore this issue (*Guo et al., 2002*; *Miao et al., 2012*; *Tang and Ding, 2013*; *Wang et al., 2022*).

As discussed, the three groups can be distinguished by different mandibular symphysis and tusk morphologies (*Gheerbrant and Tassy, 2009*), which indicate different feeding behaviors. Their distinct narial regions further reflect different evolutionary stages of their trunks (*Tassy, 1994*). *Choerolophodon* possesses a highly specialized mandibular symphysis (*Figure 1—figure supplement 5A and B*) for cutting horizontally growing plants and was confined to relatively close habitats. The low evolutionary level of the narial region suggests a relatively primitive or clumsy trunk in *Choerolophodon*. The feeding strategy of *Gomphotherium* was unspecialized and flexible, relying on the coordination of the enamel band of an upper tusk and the corresponding lower tusk. Previous research also suggested that *G. steinheimense*, the sister taxon of elephantoids (*Figure 1A*, *Figure 1—figure supplement 1*) from the Halamagai Fauna, fed on grasses (*Wu et al., 2018*). While FE analysis could not provide a clear suggestion for the trunk function of *Gomphotherium*, the narial evolutionary stage, which is close to living elephants, suggests a relatively flexible trunk in *Gomphotherium*, as it is phylogenetically closer to living elephants than the other two groups.

Enamel isotope results also support the idea that *Platybelodon* expanded its living habitats to more open environments, such as grasslands, more than any other bunodont elephantiforms, likely due to its distinct feeding strategy. Our FE analyses strongly indicate that the *Platybelodon* mandible is specifically suited for cutting vertically growing plants. In open environments, there are vertically growing plants, such as soft-stemmed herbs. *Platybelodon* did not survive to the Late Miocene in northern China, possibly due to a mass extinction caused by the global climatic shift known as the Tortonian Thermal Maximum (*Westerhold et al., 2020*). However, from another perspective, if the *Platybelodon* trunk functioned similarly to that of living elephants, such as pulling out herbs from the earth (*McKay, 1973*), the greatly enlarged mandibular symphysis and tusks would have become redundant.

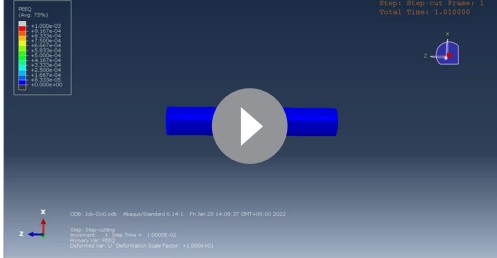

**Video 11.** Finite element (FE) modeling of *Gomphotherium* vertical twig-cutting test, the twig model (facing to the surface that is in contact with the tusk), color map showing the equivalent plastic strain.
https://elifesciences.org/articles/90908/figures#video11

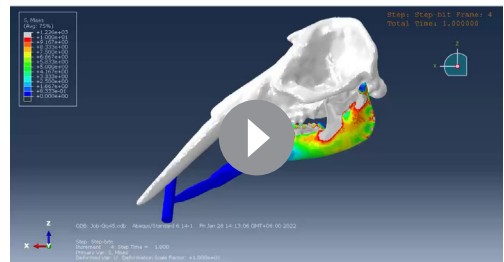

**Video 12.** Finite element (FE) modeling of *Gomphotherium* oblique twig-cutting test, the total model, color map showing the von Mises stress.
https://elifesciences.org/articles/90908/figures#video12

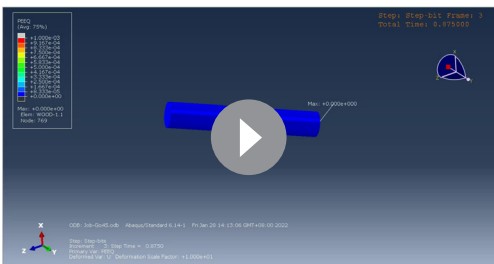

**Video 13.** Finite element (FE) modeling of *Gomphotherium* oblique twig-cutting test, the twig model (facing to the surface that is in contact with the tusk), color map showing the equivalent plastic strain. https://elifesciences.org/articles/90908/figures#video13

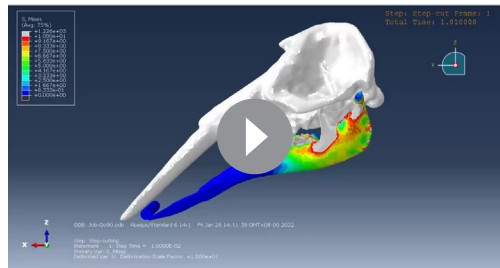

**Video 14.** Finite element (FE) modeling of *Gomphotherium* horizontal twig-cutting test, the total model, color map showing the von Mises stress. https://elifesciences.org/articles/90908/figures#video14

The reduction of the mandibular symphysis and tusks in the Late Miocene occurred in every lineage of elephantiform, coinciding with the large-scale expansion of $C_4$ grasses in the middle and low latitudes (*Cerling et al., 1997*). This may reflect the functional evolution of trunk grasping and manipulation in all elephantiforms lineages. From around 8 to 5 Ma in Africa, some derived members of 'Gomphotheriidae' (e.g. *Anancus*) and stem taxa of Elephantidae (e.g. *Stegotetrabelodon*, *Primelephas*) showed a strong inclination toward grazing on $C_4$ grasses, even though their cheek tooth morphology remained relatively primitive (*Lister, 2013*). Thus, open environments might be a key factor in both trunk development and the evolution of modern elephants.

As we have discussed, mandibular elongation was a prerequisite for the extremely long trunk of proboscideans, and open-land grazing further promoted the evolution of trunks with complex manipulative functions. This may explain why tapirs never developed a trunk as dexterous as that of elephants, as tapirs never shifted their adaptation zones to open lands. Living in dense forests, foods were easily accessible and procured through the mouth. Furthermore, the living mammals of the Late Cenozoic, in various open areas, have undergone specific organ evolution, such as the elongated necks of giraffes (*Wang et al., 2022*), the extravagate saber-tooth evolution in carnivores (*Jiangzuo et al., 2023*), and the development of various strange cranial appendages in different ruminants (*Janis, 1982*). It is possible that the highly evolved trunks of elephants evolved somewhat accidentally, under the pressure of ecological changes from closed to open environments.

## Conclusion

In this study, we have examined the functional and eco-morphology, as well as the feeding behaviors, of longirostrine bunodont elephantiforms. Our findings demonstrate that multiple eco-adaptations have contributed to the diverse mandibular morphology observed in proboscideans, while open-land grazing has driven the development of trunk coiling and grasping functions and ultimately led to the loss of the long mandible. Specifically, the longirostrine elephantiform, *Platybelodon*, represents

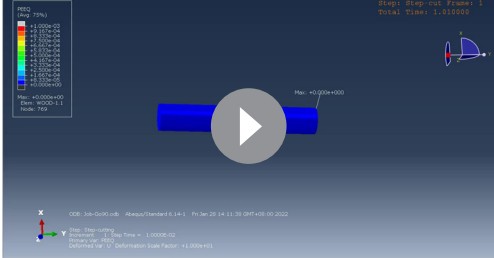

**Video 15.** Finite element (FE) modeling of *Gomphotherium* horizontal twig-cutting test, the twig model (facing to the surface that is in contact with the tusk), color map showing the equivalent plastic strain. https://elifesciences.org/articles/90908/figures#video15

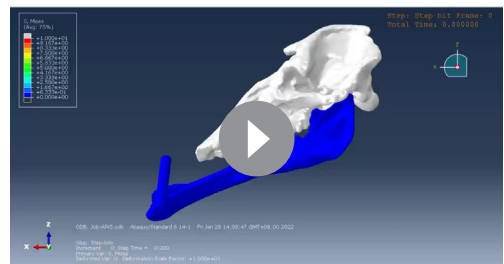

**Video 16.** Finite element (FE) modeling of *Choerolophodon* oblique twig-cutting test, the total model, color map showing the von Mises stress. https://elifesciences.org/articles/90908/figures#video16

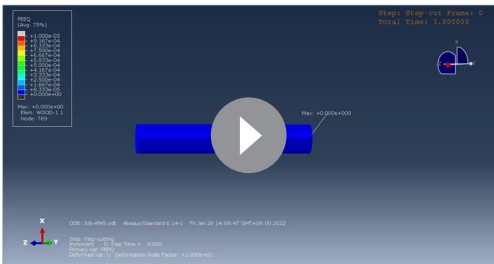

**Video 17.** Finite element (FE) modeling of *Choerolophodon* oblique twig-cutting test, the twig model (facing to the surface that is in contact with the cutting plate), color map showing the equivalent plastic strain.
https://elifesciences.org/articles/90908/figures#video17

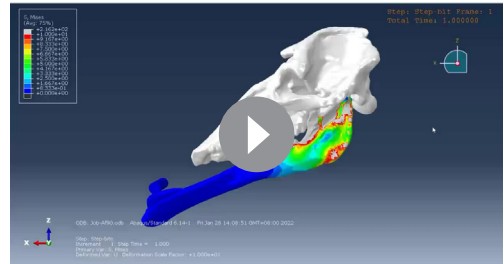

**Video 18.** Finite element (FE) modeling of *Choerolophodon* horizontal twig-cutting test, the total model, color map showing the von Mises stress.
https://elifesciences.org/articles/90908/figures#video18

the first known proboscidean to have evolved both grazing behavior and trunk coiling and grasping functions. We have arrived at this conclusion through three lines of evidence, including the palaeo-ecological reconstruction based on tooth enamel stable isotope data, the reconstruction of feeding behaviors through FE analyses, and the examination of mandibular and narial region morphology correlated with characteristics associated with horizontal cutting behavior. The coiling and grasping ability of the trunk in *Platybelodon* evolved within the ecological context of Central Asia, which experienced regional drying and the expansion of open ecosystems following the MMCT (*Miao et al., 2012*; *Tang and Ding, 2013*). As a result, *Platybelodon* outcompeted other longirostrine bunodont elephantiforms and flourished in the open environment of northern China until the end of the MMCT. This scenario sheds light on how proboscideans overcame an evolutionary bottleneck. Initially, the elongation of mandibular symphysis and tusks served as the primary feeding organs, with the trunk being used as an auxiliary tool. However, through some necessary modifications such as tactile specialization and water intake adaptations (*Shoshani, 1998*; *Purkart et al., 2022*), the feeding function gradually shifted entirely to the trunk, which offered advantages in terms of flexibility and lighter weight of the feeding organs. Consequently, elephantiforms rapidly reduced the length of their mandibular symphysis and tusks (*Shoshani and Tassy, 2005*; *Gheerbrant and Tassy, 2009*). Similar stories of open-land adaptation and the acquisition of iconic characteristics have been observed in various mega-mammalian lineages, highlighting the crucial role of open-land adaptation for successful survival in modern ecosystems (*Janis, 1982*).

## Materials and methods
### Materials
The materials examined in this work are from three longirostrine gomphothere families, i.e., Choerolophodontidae, Amebelodontidae, and 'Gomphotheriidae' (*Gheerbrant and Tassy, 2009*). These materials include complete crania, mandibles, and teeth of different species, including *C. chioticus*, *C. connexus*, and *C. guangheensis* (Choerolophodontidae); *P. dangheensis*, *P. grangeri*, *P. tetralophus*, *P. tongxinensis*, *P. brevirostris*, *Protanancus* sp., *P. wimani*, and *A. zhaoi* (Amebelodontidae); *G.* cf. *angustidens*, *G. cooperi*, *G. inopinatum*, *G. steinheimense*, *G. tassyi* ('Gomphotheriidae'). All were housed in three museums: the Institute of Vertebrate Paleontology and Paleoanthropology (IVPP), American Museum of Natural History (AMNH), and Hezheng Paleozoological Museum (HPM; HMV is the specimen prefix). For the detailed specimen list, please see *Figure 2—source data 1*. These materials were discovered from four regions, the

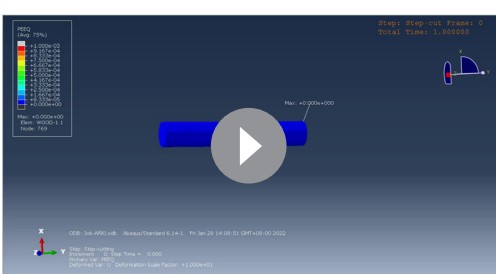

**Video 19.** Finite element (FE) modeling of *Choerolophodon* horizontal twig-cutting test, the twig model (facing to the surface that is in contact with the cutting plate), color map showing the equivalent plastic strain.
https://elifesciences.org/articles/90908/figures#video19

Linxia Basin, Tongxin region, Junggar Basin, and Tunggur region (*Figure 2*; *Figure 2—figure supplement 2*); these regions are fossil-rich, especially during the Shanwangian and Tunggurian stages (~20–11 Ma), with different fossil assemblages in different ages (*Deng et al., 2013*; *Wang et al., 2016*; *Wang et al., 2022*; *Qiu et al., 2013*). For the age information of each fossil assemblage (*Wang et al., 2022*; *Qiu et al., 2013*; *Wang, 2021*), please see *Supplementary file 1*.

## Cladistic analysis

Cladistic analyses were performed to evaluate the phylogenetic hypothesis of trilophodont longirostrine proboscideans. The data matrix contained 37 taxa, including most of the known trilophodont longirostrine taxa at the species level, and *P. serridens*, an Oligocene basal elephantiform, was selected as outgroup. Additionally, a basal elephantoid, *T. longirostris,* was also included to assess which clade the true elephantids originated from. The morphological characters included 5 characters from upper tusks, 9 from mandibular tusks, 37 from cheek teeth, 19 from the cranium, and 10 from the mandible, mainly following *Tassy, 1996*; *Shoshani, 1996 Wang et al., 2017b*. For a description of the characters and states, please see Appendix; for the data matrix, please see *Figure 1—source data 1*. Two methods, Bayesian tip-dating (BTD) and maximum parsimony (MP) analyses, were performed.

In BTD analysis, the fossil ages were incorporated as tip calibrations (*Gavryushkina et al., 2014*; *Ronquist et al., 2012*; *Zhang et al., 2016*). The Lewis Mkv model (*Lewis, 2001*), with gamma rate variation across characters (Mkv+G) (*Yang, 1994*), was initially used; subsequently, the timetree was modeled by the fossilized birth death process (*Heath et al., 2014*; *Stadler, 2010*). The process was conducted using the time of the most recent common ancestor (root age) and included hyperparameters of speciation rate, extinction rate, fossil-sampling rate, and extant-sampling probability. The root age was first assigned an offset exponential, with mean age of 37 Ma and minimum age of 34 Ma, referring to the oldest fossil. The fossil ages were fixed to their first occurrence. The extant-sampling probability was fixed to 1 because no living genera were specified. Apart from the timetree, the other key component was the relaxed clock model, which models the evolutionary rate variation along the branches in the tree. We used the independent gamma rate clock model (*Lepage et al., 2007*), in which the mean clock rate was initially assigned a lognormal prior (–6, 1) and the variance parameter of the clock rate was exponential (10).

We executed two independent runs and four chains per run (one cold chain and three hot chains) using Markov chain Monte Carlo. Each run was executed for 1 million generations and sampled every 5000 generations. The first 25% of the samples were discarded as burn-in and the rest of the two runs were combined. Good convergence and mixing were determined by effective sample sizes larger than 200 for all parameters and average standard deviations of split frequencies smaller than 0.01 (*Geyer, 1992*). The BTD analysis was performed in MrBayes 3.2.7 (*Ronquist et al., 2012*).

MP reconstruction was performed by TNT1.1 (*Goloboff et al., 2008*). In MP analysis, all characters were equally weighted. Characters 20–25 pertained to the loph/lophid numbers of cheek teeth (character numbers begin from '1' in our numeration; however, in MP analysis, TNT1.1 automatically numbered characters from '0'; therefore, in the TNT1.1 program, these characters were numbered '19–24'). These were treated as ordered and irreversible, which was performed by setting 'step-matrix → of costs' under the menu 'Data → Character settings', assigning the value '9' to the blanks where $i>j$ ($i$ to $j$ in the matrix), and assigning the value '$j$–$i$' to the blanks where $i<j$. The traditional search strategy was performed, and the results were reported based on a 50% majority consensus tree of the most parsimonious trees. Node supports were calculated by symmetric resampling with 0.33 change probability (1000 replicates). The major consensus tree was then calibrated using the time PaleoPhy function in R package paleotree 3.3.25 (*Bapst, 2012*), and is shown in *Figure 1—figure supplement 1*.

## Stable carbon and oxygen isotope analysis

In this study, 83 tooth enamel samples from three gomphothere families were collected from gomphothere specimens (*Figure 2—source data 2*) for stable carbon and oxygen isotope analysis. We also compiled previously published isotope data of *Platybelodon* from the Tunggur region and Laogou, Linxia Basin (*Wang and Deng, 2005*; *Zhang et al., 2009*). The fossil teeth used in this study are well preserved and showed no visible signs of alteration. Tooth enamel samples were obtained by cutting a small patch of enamel from a tooth or drilling along the entire length using a rotary drill; then, the samples were ground into fine powder. All samples (2–3 mg enamel powder) were then

pre-treated with 5% sodium hypochlorite (NaOCl) overnight to remove any possible organic contaminants and then cleaned with distilled water. These samples were then treated with 1 mol acetic acid overnight to remove non-structural carbonates and subsequently cleaned with distilled water. The treated samples were then freeze-dried.

The dried enamel samples were reacted with 100% phosphoric acid ($H_3PO_4$) at 25°C for approximately 72 hr. Carbon and oxygen isotope data were measured at Florida State University using a Finnigan MAT Scientific Delta Plus XP stable isotope ratio mass spectrometer coupled with a Thermo Scientific GasBench II. The lab standards that we used include MERK, MBCC, ROY-CC, and PDA. Results are reported in the standard delta ($\delta$) notation as $\delta^{13}C$ and $\delta^{18}O$ values in reference to the international carbonate standard VPDB (Vienna Pee Dee Belemnite). We reconstructed the diet $\delta^{13}C$ values of proboscideans from enamel $\delta^{13}C$ values using an enrichment factor ($\varepsilon^*$) of 13‰ for non-ruminants, like proboscideans (*Cerling and Harris, 1999*; *Passey et al., 2005*). Detailed results and specimen information are shown in *Figure 2—source data 2*.

## FE analysis

We investigated the feeding behaviors of the three longirostrine gomphothere families, i.e., Choerolophodontidae, Amebelodontidae, and 'Gomphotheriidae', using FE stimulation. Three species were selected to represent each family, *C. chioticus*, IVPP V23457 (cranium and mandible); *P. grangeri*, HMV 0930 (cranium and mandible); and *G. tassyi*, IVPP V22780 (cranium) and IVPP V22781 (mandible), respectively. Note that, in *Gomphotherium*, we were unable to get access to cranium and mandible that belonged to one individual. We used a handheld Artec Spider 3D scanner to obtain the surface topology of these specimens. The surface meshes were produced using Artec Studio 14 Professional. These meshes were first repaired with ZBrush 2021; for example, ZBrush 2021 was used to recover the broken edges of mandibular tusks, create the remaining mandibular tusks in the alveolus, create the keratinous cutting plate of the mandible in *Choerolophodon*, and for retro-deformation of the crushed cranium (i.e. the cranium of IVPP V23457).

The rough surface meshes were edited using Materialise 3-matic Research (V12.0). The surface meshes were smoothed by removing the small knobs and filling the holes for volume mesh generation. Before volume meshing, the models were cut into symmetric halves along the median sagittal plane. To reduce computation, only the right halves were preserved for further analyses. Note that the cranium was only used to define the attachments or insertions of jaw-closing muscles (modeled by many draglines, see below), and was treated as a rigid body in simulation (therefore, the cranial and nasal cavities are not relevant). The mandible contains two parts, the bony structure, including the cheek teeth, and the food acquisition organs (the mandibular tusk in *Platybelodon* and *Gomphotherium*, and the keratinous cutting plate in *Choerolophodon*). These two parts were integrated using the command 'Non-manifold Assembly' in Materialise 3-matic Research. Then, the cranium and the mandible were aligned based on their natural position, i.e., the occlusal surface of the upper and lower tooth rows were matched, and the mandibular condyle and glenoid fossa were fitted. The sagittal surfaces of the cranium and mandible were made to coincide with the $x$–$z$ plane, and the $x$-positive direction was set along the rostral direction. For comparison between different taxa, the three mandibular models were scaled to the same volume as *Choerolophodon* (6,563,708.0146 mm$^3$) (*Figure 3—figure supplement 1*). Finally, volume meshes were generated in the cranium and mandible across the three models, and exported as .inp files that could be loaded into Abaqus CAE (V6.14), the engineering software for FE analysis. The parameters for volume mesh generation are listed in *Supplementary file 1*.

The volume meshes, representing the geometric models, were imported into Abaqus CAE (V6.14) (*Figure 3—figure supplements 2–4*), and included two parts, cranium and mandible; additionally, a third part was created by Abaqus CAE, a long cylinder (300 mm in length, 50 mm in diameter), to model twigs that were cut by mandibular tusk or keratinous cutting plate. Note that the millimeter (mm)–ton (t)–second (s) unit system was adopted; other units included Newton (N) and million Pascal (MPa). Different materials, including bone, dentine, keratin, and wood, were assigned to the corresponding parts. The materials of bone, dentine, and keratin were treated as isotropic linear elastic materials. For the detailed parameters, see *Supplementary file 1* (*Drake et al., 2016*; *Bo and Quanshui, 2000*). However, twigs could not be treated as an isotropic linear elastic material, because the purpose of this simulation was to evaluate the food procuring efficiency of different taxa in different

working conditions. Here, the twigs were assigned as an orthotropic elastoplastic material, and the parameters (of wet red pine tree) were obtained from a wood handbook (*Risbrudt et al., 2010*).

The occlusal surfaces were coupled by two arbitrary points on the upper and lower teeth. These two points were connected by a 'beam connector', which constrained all the degrees of freedom (*df*) between the two points (*Figure 3—figure supplements 2–4*). In this way, we simulated the occlusal surfaces of the upper and lower teeth. Jaw-closing muscles were simulated by several groups of 'axis connectors'. This type of connector does not constrain any *df* of the two extreme points, and allows exerting force along the connector (*Figure 3—figure supplements 2–4*). Four jaw-closing muscles were considered, including *temporalis*, *superficial masseter*, *zygomaticomandibularis*, and *pterygoideus internus*. Ten axis connectors were assigned to the *temporalis*, and four, three, and three were assigned to the latter three, respectively. These connectors were uniformly arranged along their insertion areas based on their natural anatomy. The areas of the temporal fossa (*At*) and ascending ramus (*AA*) were measured in Materialise 3-matic Research (V12.0) (*Figure 3—figure supplement 1*). We estimated the muscle force of *temporalis* as follows *Tseng et al., 2017*:

$At$ (mm$^2$)×0.3
this force was equally distributed to the 10-axis connectors for temporalis.
Alternatively,
$AA$ (mm$^2$)×0.3

was considered the gross force for *superficial masseter*, *zygomaticomandibularis*, and *pterygoideus internus*. This force was also equally distributed to the other 10-axis connectors.

Note that the *At* and *AA* in the *Platybelodon* and *Gomphotherium* models were not true. These models were scaled to the same volume as that of *Choerolophodon*. In the simulation, we uniformly assigned muscle forces to make it easy to compare models (*Supplementary file 1*).

The cranium was treated as a rigid body and was fixed. Another boundary condition was assigned to a node of the mandibular condyle, which only allowed the y-direction rotation and constrained any other *df*s (simulating the rotation of the mandibular articulation). The *df*s of the x- and z-rotations of the mid-symphysis were also constrained by considering the connection to the other half.

Two tests were carried out on the composite models: the distal forces test (*dft*) and twig-cutting test (*tct*). The *dft* includes two steps: (1) applying the muscle force; and (2) exerting a distal 5000 N force that gradually changes from horizontally to vertically, by which we assessed the optimum direction of the external force for the mandible of each taxon. In this test, the 'twig' was not included in the model.

In the *tct*, the middle point of the 'twig' was set in close contact with the distal edge of the mandibular tusk and keratinous cutting plate, and was placed horizontally (*Figure 3—figure supplement 2*), 45° obliquely (*Figure 3—figure supplement 3*), and vertically (*Figure 3—figure supplement 4*). One extremity of the 'twig' was fixed, and contact properties were assigned (hard contact normally and 0.3 frictional coefficient tangentially). The *tct* also includes two steps: (1) applying the muscle force as in *dft*; and (2) displacing the cranium and mandible 10 mm toward the 'twig' to stimulate cutting action of proboscideans, by which we determined the cutting efficiency of directions for each taxon. In the results, the SEPS from total twig elements was calculated and reported for each model. The plastic strain represents the irreversible deformation of an element, and the SEPS from all twig elements can reflect the cutting effects in each model. The videos for von Mises stress contour color maps were also generated in the *dft* (*Videos 1–3*) and *tct* (*Video 4*, *Video 6*, *Video 8*, *Video 10*, *Video 12*, *Video 14*, *Video 16*, *Video 18*) modeling, and for the EPS contour color maps of twigs in *tct* (*Video 5*, *Video 7*, *Video 9*, *Video 11*, *Video 13*, *Video 15*, *Video 17*, *Video 19*) modeling.

## PCA and PC scores mapping on the tree

In PCA, only taxa of the three gomphothere families—members of Choerolophodontidae, Amebelodontidae, and 'Gomphotheriidae', in addition to *P. serridens*—were retained; the mammutid taxa (*Losodokodon*, *Eozygodon*, and *Zygolophodon*) and the stem elephantimorphs (*Eritreum* and *Gomphotherium annectens*) were excluded from the analyses.

A character combine, including characters 1–14 of upper and lower tusks, as well as 71–73, 76, 77, and 80 of the mandibles (Supplementary appendix and *Figure 1—source data 1*), was generated using PCA, which represents the synthetic character states of food acquisition organs. Besides,

another two character combines, four characters concerning the narial region (i.e. characters 54–57) and five characters (characters 5, 9, 11, 72, and 77) in relation to the horizontal cutting behavior, were also generated using PCA. PCA was performed in Past 4.04, in which the missing values were predicted using mean value imputation. The data on PC1 vs. PC3 or PC2 vs PC3 plans were plotted (*Figure 1—figure supplement 5A, C, D*).

The PC1 of the food acquisition organ combination and that of the narial region, as well as the PC2 of the character combine in relation to the horizontal cutting behavior were selected to represent the synthetic evolutionary state of each (*Figure 1—figure supplement 4*; *Figure 1—figure supplement 5B*). We use PC2 rather than PC1 in the last test because the characters in relation to the horizontal cutting behavior were not evolved in one way. For example, the character 9 indicates the width of the mandibular tusks, which includes three states: 0, wide; 1, very wide; 2, narrow. In this case, the larger value (2=narrow mandibular tusker) does not mean the stronger horizontal cutting effect. Finally, these PCs were respectively mapped on the BTD tree (mammutids and stem elephantimorphs removed), using the contMap function in R package phytools 0.7-90 (*Bapst, 2012*).

## Acknowledgements

We thank the financial support received during the field work of the Second Tibetan Plateau Scientific Expedition. We appreciate Jie Ye, Xiaoxiao Zhang for his great guidance and suggestions on taxonomy and stratigraphy; Chi Zhang for assistance in phylogeny; Sijian Xu for assistance in fossil preparation; Jin Meng of AMNH, Rong Yang and Shanqin Chen of HPM for access to specimens in these two museums for comparison and statistics; Xiaocong Guo and Yu Wang for reconstruction. We thank Mallory Eckstut, PhD, from Liwen Bianji (Edanz) (https://www.liwenbianji.cn) for editing the English text of a draft of this manuscript. This work was supported by the National Key Research and Development Program of China (No. 2023YFF0804501), National Natural Science Foundation of China (52178141, 41625005), National Science Foundation Cooperative Agreement No. DMR-1157490, the State of Florida.

## Additional information

### Funding

| Funder | Grant reference number | Author |
| --- | --- | --- |
| National Natural Science Foundation of China | 52178141 | Shiqi Wang |
| National Key Research and Development Program of China | No.2023YFF0804501 | Shiqi Wang |
| National Natural Science Foundation of China | 41625005 | Shiqi Wang |

The funders had no role in study design, data collection and interpretation, or the decision to submit the work for publication.

### Author contributions

Chunxiao Li, Data curation, Software, Formal analysis, Investigation, Writing – original draft, Writing – review and editing; Tao Deng, Funding acquisition; Yang Wang, Resources, Data curation, Software, Methodology; Fajun Sun, Burt Wolff, Qigao Jiangzuo, Jiao Ma, Luda Xing, Jiao Fu, Visualization; Ji Zhang, Software, Visualization; Shiqi Wang, Resources, Software, Supervision, Methodology

### Author ORCIDs

Chunxiao Li https://orcid.org/0000-0002-9278-3473
Yang Wang http://orcid.org/0000-0002-8232-3289
Qigao Jiangzuo http://orcid.org/0000-0003-4773-5349
Ji Zhang https://orcid.org/0000-0001-8206-4531
Shiqi Wang https://orcid.org/0000-0001-7752-5620

Reviewer #1 (Public Review): https://doi.org/10.7554/eLife.90908.3.sa1
Author response https://doi.org/10.7554/eLife.90908.3.sa2

## Additional files

### Supplementary files

• Supplementary file 1. 1 Age estimation of different faunas of the four regions. 2 Geometric and mesh parameters of the finite element (FE) models. 3 Material properties in the finite element (FE) models. 4 Muscle forces estimation of finite element (FE) models.

• Supplementary file 2. 3D model 1 *Platybelodon* cranium model (.stl file), 3D model 2 *Platybelodon* mandible model (.stl file), 3D model 3 *Gomphotherium* cranium model (.stl file), 3D model 4 *Gomphotherium* mandible model (.stl file), 3D model 5 *Choerolophodon* cranium model (.stl file), 3D model 6 *Choerolophodon* mandible model (.stl file).

• MDAR checklist

### Data availability

All data generated or analysed during this study are included in the manuscript and supporting files; Codes' files have been uploaded on the Dryad Dataset platform.

The following dataset was generated:

| Author(s) | Year | Dataset title | Dataset URL | Database and Identifier |
|---|---|---|---|---|
| Li C, Wang S | 2023 | Supplementary codes | https://doi.org/ 10.5061/dryad. d2547d86g | Dryad, 10.5061/dryad. d2547d86g |

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

## Appendix 1

### Morphological characters in the phylogenetic analyses

Upper tusk

1. Thickness, in adult male: thick (0); thin (1); absent (2); very thick (3).
2. Bend: ventrally bent (0); straight (1); dorsally bent (2).
3. Spirality: not spiral (0); slightly spiral (1); strongly spiral (2).
4. Diverge of the two tusks: moderately diverge (0); parallel (1); strongly diverge (2).
5. Enamel: with an enamel band (0); without enamel (1); enamel enclosing the entirely tusk (2).

Lower tusk

1. Presence: present (0); absent (1).
2. Bend: dorsally bent (0); straight (1).
3. Spirality: slightly spiral (0); straight (1).
4. Width: wide (0); very wide (1); narrow (2).
5. Enamel: without enamel (0); with an enamel band (1).
6. Applanation of cross-section: flattened (0); extremely flattened (1); pyriform (2); round (3).
7. Median edge: straight (0); round (1).
8. Structure of cross-section: concentric laminae (0); dentine rods (1).
9. Length beyond alveolus: long (0); short (1).

Cheek teeth

1. DP2: possessing a large anterior cone (0); possessing a weak anterior cone (1).
2. DP3 and dp3: possessing only two lophs/lophids (0); possessing two lophs/lophids plus a strong posterior cingulum/cingulid (1); possessing three lophs/lophids (2).
3. DP3 and dp3, second loph/lophid: straight (0); oblique (1).
4. DP3: cones aligned (0); cones alternatively arranged (1).
5. dp3: conids aligned (0); conids alternatively arranged (1).
6. Dp4/dp4: possessing two lophs/lophids plus a strong posterior cingulum/cingulid (0); possessing three lophs/lophids (1); possessing four lophs/lophids (2).
7. Premolars: p2 present (0); p3 present (1); p4 present (2); p4 absent (3).
8. M1/m1: trilophodont (0), tetralophodont (1).
9. M2/m2: trilophodont (0), tetralophodont (1).
10. M3: 3rd loph incomplete (0); 3rd loph complete (1); 4th loph incomplete (2); 4th loph complete (3); pantalophodont (4).
11. m3: trilophodont (0); 4th lophid incomplete (1); 4th lophid complete (2); pantalophodont (3); hexalophodont (4).
12. Upper molars, the 2nd loph: posterior pretrite central conule weak or absent (0); posterior pretrite central conule well-developed (1).
13. Upper molars, the 2nd loph: posterior pretrite central conule smaller than, at most equivalent to the anterior pretrite central conule (0); posterior pretrite central conule larger than the anterior pretrite central conule.
14. Lower molars, the 2nd lophid: anterior pretrite central conule weak or absent (0); anterior pretrite central conule well developed (1).
15. Lower molars, the 2nd lophid: posterior pretrite central conule not enlarged (0); posterior pretrite central conule enlarged (1).
16. Molars, typically in the 2nd loph/lophid: crescentoids absent (0); crescentoids weak (1); crescentoids complete (2).
17. Molars, typically in the 2nd loph/lophid: posttrite mesoconelet(s) well developed (0); posttrite mesoconelet(s) weak or absent (1).
18. Molars, typically in the 2nd loph/lophid: pretrite mesoconelet individualized (0); pretrite mesoconelet fused with the anterior pretrite central conule (1).

19. Molars, typically in the 2nd loph/lophid: posttrite central conule(s) absent (0); posttrite central conule(s) present (1).
20. Upper molars, typically in the 2nd loph: posttrite half loph divided into two to three elements (main conelet and mesoconelet(s)) (0); posttrite half loph subdivided more than three elements (2); posttrite half loph subdivided as a thin crest (3).
21. Molars: elements of posttrite half lophs/lophids inflated (0); posttrite half lophs/lophids compressed (1); posttrite half lophs/lophids highly compressed as a thin crest (2).
22. Molars, typically in the 2nd loph/lophid: each pretrite central conule composed of 1–2 conules (0); each pretrite central conule composed of more than 3 conules, showing a thick crest (1); pretrite central conule absent (2).
23. Upper molars, zygodont crest: absent (0); thick (1); thin (2).
24. Lower molars, zygodont crest: absent (0); present (1).
25. Molars, typically in the 2nd loph/lophid: pretrite trefoil incomplete (0); pretrite trefoil well developed (1); pretrite trefoil with thin, crest-like lobes (2); pretrite trefoil secondary weakened or at least showing the tendency (3).
26. Molars, typically in the 2nd loph/lophid: posttrite trefoil absent (0); posttrite trefoil incomplete (1); posttrite trefoil complete (2).
27. Upper molars, typically in the 2nd loph: pretrite and posttrite half lophs aligned (0); pretrite and posttrite half lophs more or less chevroned (1).
28. Lower molars, typically in the 2nd lophid: pretrite and posttrite half lophids aligned (0); pretrite and posttrite half lophids more or less chevroned (1).
29. Lower molars, typically in the 2nd lophid: posttrite half lophid normal to mid-axis (0); posttrite half lophid distadaxially oblique (1).
30. Molars: without anancoid contact (0); with pseudanancoid contact (1).
31. Upper molars, typically in the 1st interloph: entoflexus 'I-shaped' (compressed) (0); entoflexus 'V-shaped' (1); entoflexus open (2).
32. Lower molars, typically in the 1st interlophid: ectoflexid 'V-shaped' (compressed) (0); ectoflexid 'U-shaped' (1); ectoflexid open (2); ectoflexid 'I-shaped' (highly compressed) (3).
33. Molars, width: in normal width (0); narrow (1); broad (2).
34. Molars, ptychodonty: absent (0); present (1).
35. Molars, choerodonty: absent (0); present (1).
36. Molars, cementodonty: absent (0); present (1).
37. Molars, crown height: low crowned (0); slightly high crowned (1).

## Cranium

1. Braincase: narrow (0); wide (1).
2. Braincase: low (0); relatively high (1).
3. Narial, nasal aperture: rostral to the orbit (0); upon the orbit (1).
4. Narial, perinasal fossa: absent or weak (0); with a complete perinasal fossa (1); extremely large, showing a prenasal slope (2).
5. Nasal bone, nasal process: large (0); small (1).
6. Mesethmoid cartilage insertion: large (0); small (1).
7. Subnasal fossa: absent (0); present (having an incisive constriction) (1).
8. Rostrum: narrow (0); wide (1).
9. Rostrum: short (0); long (1).
10. Lachrymal opening: present (0); absent (1).
11. Orbital: in normal position (0); caudally positioned (1); dorsally positioned (2); rostrally positioned (1).
12. Orbitotemporal crest: oblique (0); vertical (1).
13. Facial region: rostrally positioned (0) caudally positioned (1); ventrally stretched (2).
14. Post palatine: with a spine (0); with a tuberosity (1); without ornamentation (2).
15. Basicranium: not erected (0); erected (1).
16. Zygomatic processes: wide in distance (0); narrow in distance (1).
17. Tympanic: narrow (0); inflated (1).

18. Tympanic: separated foramen lacerum medium and foramen ovale (1); merged foramen of the above.
19. Tympanic: internal carotid artery foramen large (0); internal carotid artery foramen small or absent (1).

## Mandible

1. Symphysis: long (0); short (1); extremely elongated (2).
2. Symphysis: wide (0); narrow (1); extremely widened (2).
3. Symphysis: shallow (0); deep (1).
4. Symphysis: with keratinous structure (0); without keratinous structure (1).
5. Symphysis: with a transverse ledge at the proximal end (0); without a transverse ledge at the proximal end (1).
6. Symphysis: not ventrally bent (0); ventrally bent (1).
7. Symphysis: proximal end close to the tooth row (0); that distant from the tooth row (1).
8. Angular process: not elevated (0); elevated (1).
9. Angular process: posteriorly protruded (0); not posteriorly protruded (1).
10. Ramus: vertical (0); caudally inclined (1).

