## [Editor Report · eLife assessment]

This study presents **fundamental** findings on the evolution of extremely elongated mandibular symphysis and tusks in longirostrine gomphotheres from the Early and Middle Miocene of northern China. The integration of multiple methods provides **compelling** results in the eco-morphology, behavioral ecology, and co-evolutionary biology of these taxa. In doing so, the authors elucidate the diversification of fossil proboscideans and their likely evolutionary responses to late Cenozoic global climatic changes.

---

## [Referee Report · Reviewer #1 (Public Review)]

Summary:

The authors were trying to understand the relation between the development of large trunks and longirrostrine mandibles in bunodont proboscideans of Miocene, and how it reflects the variation in diet patterns.

Strengths:

The study is very well supported, written, and illustrated, with plenty Supplementary materials. The authors included all Asian bunodont proboscideans with long mandibles and I suggest that they should use the expression "bunodont proboscideans" instead of gomphotheres.

Weaknesses:

I believe that the only weakness is the lack of discussion comparing their results with the development of gigantism and long limbs in proboscideans from the same epoch.

The authors reviewed the manuscript according to my suggestions and responded well to all my comments.

---

## [Author Response]

The following is the authors’ response to the original reviews.

**Reviewer #1 (Public Review):**
Summary:The authors were trying to understand the relationship between the development of large trunks and longirrostrine mandibles in bunodont proboscideans of Miocene, and how it reflects the variation in diet patterns.Strengths:The study is very well supported, written, and illustrated, with plenty of supplementary material. The findings are highly significant for the understanding of the diversification of bunodont proboscideans in Asia during Miocene, as well as explaining the cranial/jaw disparity of fossil lineages. This work elucidates the diversification of paleobiological aspects of fossil proboscideans and their evolutionary response to open environments in the Neogene using several methods. The authors included all Asian bunodont proboscideans with long mandibles and I suggest that they should use the expression "bunodont proboscideans" instead of gomphotheres.Weaknesses:I believe that the only weakness is the lack of discussion comparing their results with the development of gigantism and long limbs in proboscideans from the same epoch.

Thank you for your comprehensive review and positive feedback on our study regarding the co-evolution of feeding organs in bunodont proboscideans during the Miocene. We appreciate your suggestion, and have decided to use the term "bunodont elephantiforms" (for more explicit clarification, we use elephantiforms to exclude some early proboscideans, like Moeritherium, ect.) instead of "gomphotheres," and we will make this change in our revised manuscript. We also appreciate the potential weakness you mentioned regarding the lack of discussion comparing our results with the development of gigantism and long limbs in proboscideans from the same epoch. We agree with the reviewer’s suggestion, and we are aware that gigantism and long limbs are potential factors for trunk development. Gigantism resulted in the loss of flexibility in elephantiforms, and long limbs made it more challenging for them to reach the ground. A long trunk serves as compensation for these limitations. limb bones were rare to find in our material, especially those preserved in association with the skull.

**Reviewer #2 (Public Review):**
This study focuses on the eco-morphology, the feeding behaviors, and the co-evolution of feeding organs of longirostrine gomphotheres (Amebelodontidae, Choerolophodontidae, and Gomphotheriidae) which are characterised by their distinctive mandible and mandible tusk morphologies. They also have different evolutionary stages of food acquisition organs which may have co-evolve with extremely elongated mandibular symphysis and tusks. Although these three longirostrine gomphothere families were widely distributed in Northern China in the Early-Middle Miocene, the relative abundances and the distribution of these groups were different through time as a result of the climatic changes and ecosysytems.These three groups have different feeding behaviors indicated by different mandibular symphysis and tusk morphologies. Additionally, they have different evolutionary stages of trunks which are reflected by the narial region morphology. To be able to construct the feeding behavior and the relation between the mandible and the trunk of early elephantiformes, the authors examined the crania and mandibles of these three groups from the Early and Middle Miocene of northern China from three different museums and also made different analyses.The analyses made in the study are:(1) Finite Element (FE) analysis: They conducted two kinds of tests: the distal forces test, and the twig-cutting test. With the distal forces test, advantageous and disadvantageous mechanical performances under distal vertical and horizontal external forces of each group are established. With the twig-cutting test, a cylindrical twig model of orthotropic elastoplasity was posed in three directions to the distal end of the mandibular task to calculate the sum of the equivalent plastic strain (SEPS). It is indicated that all three groups have different mandible specializations for cutting plants.(2) Phylogenetic reconstruction: These groups have different narial region morphology, and in connection with this, have different stages of trunk evolution. The phylogenetic tree shows the degree of specialization of the narial morphology. And narial region evolutionary level is correlated with that of character-combine in relation to horizontal cutting. In the trilophodont longirostrine gomphotheres, co-evolution between the narial region and horizontal cutting behaviour is strongly suggested.(3) Enamel isotopes analysis: The results of stable isotope analysis indicate an open environment with a diverse range of habitats and that the niches of these groups overlapped without obvious differentiation.The analysis shows that different eco-adaptations have led to the diverse mandibular morphology and open-land grazing has driven the development of trunk-specific functions and loss of the long mandible. This conclusion has been achieved with evidence on palaecological reconstruction, the reconstruction of feeding behaviors, and the examination of mandibular and narial region morphology from the detailed analysis during the study.All of the analyses are explained in detail in the supplementary files. The 3D models and movies in the supplementary files are detailed and understandable and explain the conclusion. The conclusions of the study are well supported by data.

We appreciate your detailed and insightful review of our study. Your summary accurately captures the essence of our research, and we are pleased to note that multiple research methods were used to demonstrate our conclusions. Your recognition of the evidence-based conclusions from paleoecological, feeding behavior reconstruction, and morphological analyses reinforces the validity of our findings. Once again, we appreciate your time and thoughtful reviews.

**Reviewer #1 (Recommendations For The Authors):**
Thank you very much for the invitation to review this amazing manuscript. It is very well written and supported, and I have only minor suggestions to improve the text:(1) Some references are not in chronological sequence in the text, and this should be reviewed.

We greatly appreciate the positive comments of the reviewer. We revised the reference of the manuscript as the reviewer’s suggestion.

(2) I suggest the use of the expression "bunodont proboscideans" instead of Gomphotheres because there is no agreement if Amebelodontidae and Choerolophodontidae are within Gomphotheriidae, as well as some brevirrostrine bunodont proboscideans from South America. So I think it is ok to use "Gomphotheriidae", but not gomphotheres to refer to all bunodont proboscideans included in the study.

The reviewer is correct. Using “gomphotheres” to refer to these three groups is inappropriate. We have replaced “gomphotheres” with "bunodont elephantiforms" throughout the entire manuscript. Here, we use “elephantiforms”, not “proboscideans”, to avoid confusion with some early proboscidean members like Moeritherium, ect.

(3) I was expecting some discussion on the development of large trunks related to the gigantism in these bunodont proboscideans, regarding the huge skulls and the columnar limbs.

We appreciate this suggestion, and we are aware that gigantism is a potential factor for trunk development. It is difficult to compare the three groups (Amebelodontidae, Choerolophodontidae, and Gomphotheriidae) in terms of their weight and limb bone length, because in our material, limb bones were rarely found, especially those associated with cranial material. Nevertheless, at this stage, all elephantiforms had significantly enlarged cranial sizes and limb bone lengths compared to early members like Phiomia. Gigantism caused the loss of flexibility in elephantiforms, and even the long limbs made it more difficult for an elephantiform to reach the ground. A long trunk compensates for this evolutionary change. Exploring these aspects further is a part of our future work.

(4) The reference to Alejandro et al should be replaced by Kramarz et al (and the correct surname of the authors). The name and surname of this reference need to be corrected. The correct names are Kramarz, A., Garrido, A., Bond, M. 2019. Please correct this in the text too.

We thank the reviewer for catching this error. This reference has been corrected.

**Reviewer #2 (Recommendations For The Authors):**
I believe your paper will lead to other studies on other Proboscidean groups on the evolution of the mandible and trunk. There are some corrections in the text:In line 199 in the text in pdf, "Tassy, 1994" should be "Tassy, 1996".In line 241, "studied" should be "studies"In line 313, "," after the word "tool" should be "."

We appreciate the reviewer for pointing these errors out and have revised these based on the suggestions.

In the References, you write "et al." in some references. You should write the names of all of the authors.In the References: "Lister AM. 2013" and "Shoshani&Tassy" are not referenced in the text.In the References: "Tassy P. Gaps, parsimony, and early Miocene elephantoids (Mammalia), with a re-evaluation of Gomphotherium annectens (Matsumoto, 1925). Zool. J. Linn." should be "Tassy P. 1994. Gaps, parsimony, and early Miocene elephantoids (Mammalia), with a re-evaluation of Gomphotherium annectens (Matsumoto, 1925). Zool. J. Linn. 112, 1-2, 101-117" and replaced before "Tassy P. 1996".

We appreciate the reviewer’s suggestions and have revised these references.